# An optogenetics device with smartphone video capture to introduce neurotechnology and systems neuroscience to high school students

Liudi Luo[1], Bryce W. Hina[1☯], Brennan W. McFarland[1☯], Jillian C. Saunders[1☯], Natalie Smolin[1☯], Catherine R. von Reyn[1,2]*

1 School of Biomedical Engineering, Science and Health Systems, Drexel University, Philadelphia, Pennsylvania, United States of America, 2 Department of Neurobiology and Anatomy, Drexel University College of Medicine, Philadelphia, Pennsylvania, United States of America

☯ These authors contributed equally to this work.
* crv33@drexel.edu

**Data Availability Statement:** All relevant data are within the manuscript, Supporting Information files, and/or deposited on a GitHub repository

## Abstract

Although neurotechnology careers are on the rise, and neuroscience curriculums have significantly grown at the undergraduate and graduate levels, increasing neuroscience and neurotechnology exposure in high school curricula has been an ongoing challenge. This is due, in part, to difficulties in converting cutting-edge neuroscience research into hands-on activities that are accessible for high school students and affordable for high school educators. Here, we describe and characterize a low-cost, easy-to-construct device to enable students to record rapid *Drosophila melanogaster* (fruit fly) behaviors during optogenetics experiments. The device is generated from inexpensive Arduino kits and utilizes a smartphone for video capture, making it easy to adopt in a standard biology laboratory. We validate this device is capable of replicating optogenetics experiments performed with more sophisticated setups at leading universities and institutes. We incorporate the device into a high school neuroengineering summer workshop. We find student participation in the workshop significantly enhances their understanding of key neuroscience and neurotechnology concepts, demonstrating how this device can be utilized in high school settings and undergraduate research laboratories seeking low-cost alternatives.

## Introduction

Neurotechnology refers to any technology used to understand and/or manipulate the structure, activity, and function of the nervous system [1]. At the clinical level, it includes techniques to enhance or repair brain function and monitor brain activity in real time. But it has also expanded beyond the clinical setting to significantly alter the way we interact with each other and our environment [1–4].

Consequently, it is then no surprise that STEM career opportunities in neurotechnology are on a rising trend predicted to continue well into the future [1, 3, 5]. In recognition of this

(https://github.com/Drexel-NCE-Lab/Arduino_
Optogenetics_Workshop). This includes annotated
data and raw videos.

**Funding:** This material is based on work partially
supported by the National Science Foundation
under Grant No. IOS-1921065 and Grant No.
CBET-1747506 (PI Catherine R. von Reyn). The
funders had and will not have a role in study
design, data collection and analysis, decision to
publish, or preparation of the manuscript.

**Competing interests:** The authors have declared
that no competing interests exist.

trend, universities have significantly increased their offerings of undergraduate courses in neuroscience and neuroengineering [6]. High school curricula, however, have lagged behind [7, 8]. This represents a missed opportunity to recruit future neuroscientists and neuroengineers [7, 9–12]. This also represents a missed opportunity to have an educated public that is familiar with these technologies as they are becoming increasingly prevalent in our modern world [7–12]. Therefore, there exists a great need to expose high school students to neuroscience and neurotechnology prior to their undergraduate education.

One of the best ways to fill this need is through active learning where students participate in the learning process, instead of passively receiving and memorizing information [7, 13, 14]. Active learning has emerged as a critical component in STEM education. When employed well, active learning significantly increases students' exam scores and concept inventories [15–17]. Active learning relies on hands-on activities that follow the principles of scientific inquiry, for which hypothesis-driven laboratory experiments are well suited. However, many cutting-edge neuroscience/neurotechnology experiments are costly [18]. In recognizing this need, there have been concerted outreach efforts from higher education, the nonprofit research sector, and the private sector to develop low-cost versions of cutting-edge devices that enable high school students to employ neurotechnologies and engage in hands-on activities to learn neuroscience concepts [8, 19–27].

One such technology is optogenetics, the process of harnessing light to control the activity of genetically modified neurons [28]. Optogenetics has been fundamental for recent advances in mapping functional brain connectivity [28, 29], and may emerge as a less invasive, neural cell-type specific option for human brain stimulation, as compared to current electrode-based cortical and deep brain stimulators [30–36]. When employed in a research laboratory, however, standard optogenetics experiments rely on sophisticated stimulation devices and expensive equipment for monitoring neural activity and/or behavior that are not available to high schools [37–41]. To increase accessibility, recent efforts have succeeded in establishing significantly lower-cost optogenetics devices [19, 20, 22, 25]. However, some of these elegant solutions have been designed for undergraduate labs and/or require 3-D printing that may not exist at high schools with limited budgets.

One of the most accessible models for optogenetic investigations is the fruit fly, *Drosophila melanogaster*. In *Drosophila*, optogenetics has been used to activate neurons in order to determine their role in making specific behavioral decisions. For example, flies may choose to take-off or freeze in response to an object, like a fly swatter or a predator, approaching on a direct collision course [38, 42, 43]. Recent experiments utilizing optogenetics have been able to determine a subset of the neurons that drive takeoffs or freezing [38, 41, 44, 45]. When translated to an educational setting, these optogenetics experiments provide an opportunity to introduce technologies for neural control. They also provide active learning opportunities to introduce and teach concepts such as neural properties, neural circuits, and how circuits function to generate behavior [37]. These experiments are also ideal for replicating with high school students, as the behaviors are relatable to the maneuvers that humans perform in response to approaching objects. The fly's behavioral outputs, when recorded with high resolution/high speed videography, are highly salient and easily scored by an untrained eye. However, many low-cost optogenetics devices designed for high school students do not provide a solution for synchronized light activation and video capture [27, 37].

Here, we describe a low-cost device to enable high school students to replicate optogenetics experiments. The device utilizes an Arduino development kit and adapts a common stereoscope and smartphone into an assay capable of acquiring behavioral data from transgenic *Drosophila melanogaster*. We integrate this device into a workshop for high school students as an active learning activity. Student assessments before and after the workshop demonstrate a

significant increase in understanding neuroscience and neurotechnology concepts. Due to its practicality and low cost, we propose our device has broad applicability for high school neuroscience labs.

## Materials and methods

A complete protocol, that encompasses building the device and running the workshop, can be found at dx.doi.org/10.17504/protocols.io.b39gqr3w. All materials with purchasing information can be found in the S1 Table. All data (videos and annotations) can be accessed at https://github.com/Drexel-NCE-Lab/Arduino_Optogenetics_Workshop.

### Optogenetics device

An overview of the optogenetics device is shown in Fig 1.

**Stereoscope Camera Mount.**  A Stereoscope Camera Mount was developed to capture video data through a stereoscope's eye piece using a smartphone camera. The mount consisted of a platform and two sleeves that were hand-made from used cardboard boxes (Fig 1A and 1B). The platform secured both the smartphone and the Motion Sensor Module (see Motion Sensor Module below) for synchronized light activation and video capture. The sleeves connected the platform to the stereoscope's eyepieces and permitted z-axis adjustments to align the camera focal length with the stereoscope's focal length so that any mobile phone can take a focused image through the stereoscope eyepiece while using the mount. The schematic for the camera mount is included in the S1 Fig.

We designed our Stereoscope Camera Mount around the Leica EZ4 stereoscope, however the sleeves connecting the platform to the stereoscope eyepieces can be modified length and width wise to accommodate any stereoscope. After constructing the sleeves for the eyepiece, if further adjustments are necessary to ensure a snug fit, modeling clay can be inserted between the stereoscope and the sleeves to adjust the z-axis, or in between the sleeves and eyepieces to adjust the xy-axis. The platform dimensions should also be capable of supporting any smartphone.

### Controller

A controller was developed to synchronize light activation for optogenetics experiments and smartphone video capture to record the behavior of transgenic *Drosophila melanogaster*. The controller was built using an Arduino Uno R3 development board, and included a Status LED Module, an Optogenetic Stimulation LED Module, and a Motion Sensor Module (Fig 1A, 1D and 1E). All components, other than the optogenetic stimulation LED (627 nm, LUXEON Rebel LED, LuxeonStar), came from an ELEGOO UNO Project Super Starter Kit. The controller wiring diagram is included in the S2 Fig.

**Status LED Module.**  The Status LED Module was used to indicate the current status of the development board. It contained an RGB LED, three 1 kΩ protective resistors, and two push switches. The RGB LED was comprised of three independent LEDs with red, green, and blue occupying digital pins D6, D5, and D4 of the development board, respectively. In our device, we only used the green and blue LEDs, pulling their digital pins to HIGH or LOW to turn them on or off. All LEDs LOW (off) indicated that the development board was dormant, green HIGH (on) indicated the board was ready to run an optogenetics experiment, and blue HIGH (on) indicated an optogenetic stimulation event had occurred. The two push switches that occupied digital pins D2 and D3 of the development board were used to switch the development board into the ready state or dormant state.

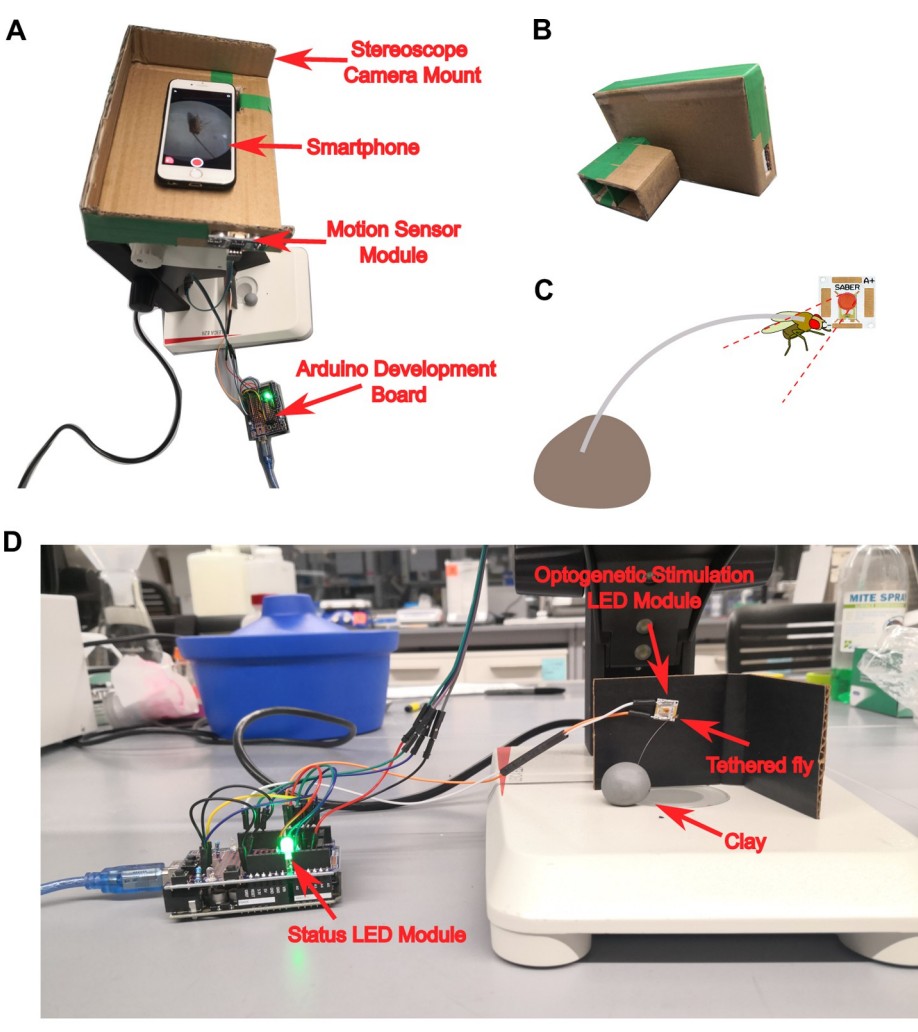

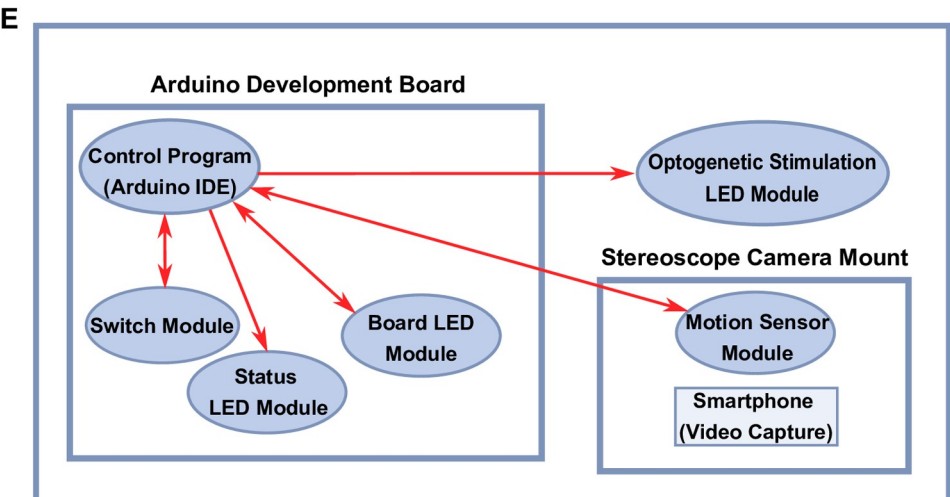

**Fig 1. A low-cost optogenetics device.** (**A**) Top view of the device mounted on a standard stereoscope. The eyepieces of the stereoscope were inserted into the sleeves of the Stereoscope Camera Mount. A smartphone was fixed in place on the Stereoscope Camera Mount, with the camera aligned to the eyepiece and the display clearly visible to the students. The Motion Sensor Module was located on the left side of the Stereoscope Camera Mount to detect finger movements. (**B**) The cardboard Stereoscope Camera Mount. Dimensions can be found within the technical drawing of

S1 Fig. (**C**) Schematic representation of a tethered *Drosophila*. UV glue connects one end of a tungsten wire to the notum of the *Drosophila*. The other end of the tungsten wire is inserted into a piece of clay for positioning and stabilization. (**D**) Side view of the device mounted on a standard stereoscope, that includes the position of the tethered *Drosophila* in front of the Optogenetic Stimulation LED Module. The Arduino Development Board depicted in this figure has a shield that replaces the role of an ordinary protoboard. (**E**) Component diagram for the device. The Arduino Development Board controls the Connected Switch Module, Status LED Module, Board LED Module, Optogenetic Stimulation LED Module, and Motion Sensor Module necessary for the experiments. Modules depicted with double-sided arrows communicate to and from the Arduino for proper module function. The Stereoscope Camera Mount houses both the Motion Sensor Module and a user's smartphone.

**Motion Sensor Module.** The Motion Sensor Module was used to detect the finger motion of the experimenter pressing the shutter button on their smartphone, and then initiate the optogenetic light stimulation. This module used an ultrasonic sensor HC-SR04 that includes ultrasonic transmitters, a receiver, and a control circuit. This module follows the principles of radar distance detection. A high frequency signal (50 kHz) was generated with a transmitter and the latency from sending a signal to receiving an echo reflected from an object was measured. The distance of the object from the transmitter was then calculated using the following equation:

$$D = \frac{T \times V}{2} \tag{1}$$

where $D$ is the distance, $T$ is the latency between ultrasonic transmission and reception, and $V$ is the speed of sound.

In our device, the ultrasonic transmitters occupied digital pin D11 as output and the receiver occupied digital pin D12 as input. By repeatedly pulling D11 to LOW for 5 μs and then switching D11 to HIGH for 10 μs, we generated 50 kHz waves for our transmitter. Based on the calculated transmission to reception latency, the distance of the object in front of the ultrasonic sensor was calculated using Eq (1). Based on the geometry of the camera mount, a distance of less than 10 cm was used as the threshold for finger motion detection.

**Optogenetic Stimulation LED Module.** The Optogenetic Stimulation LED Module was used to activate neurons within transgenic *Drosophila melanogaster* expressing a red-shifted channelrhodopsin, CsChrimson [46]. This module consisted of a red LED (627 nm, LUXEON Rebel) occupying digital pin D7 that was set to LOW as a default. When the Motion Sensor Module detected a finger press of the smartphone camera shutter, the RGB LED in the Controller LED Module changed from green to blue. After a 1.5 s delay, the level of digital pin D7 was changed from LOW to HIGH for 300 ms, causing a 300-ms activation stimulation from the red LED (measured to be 0.16 mW/mm$^2$ at the fly's location, 2 mm away from the LED). At the end of the LED stimulation, the device entered a 30 s refractory period. During this period, students would stop their video recording and the Optogenetic Stimulation LED could not be re-activated, preventing students from over-stimulating their fruit fly and providing enough time for CsChrimson to recover from deactivation [46]. The timing of the video capture and light stimulation enabled students to consistently collect fly behavior before, during and after the light stimulation.

**Software accessibility.** All controller code generated for this device, as well as open-source code used to run the device, can be accessed in the S1 File.

## Experimental procedures

***Drosophila* rearing.** *Drosophila melanogaster* were raised in the dark, in foil covered vials, on standard cornmeal/molasses medium (control flies), or as larva on standard food plus 0.2 mM all-*trans*-retinal (retinal) that was switched to standard food plus 0.4 mM retinal upon

**Table 1. Genotypes.**

| Name | Full Genotype Name | References |
|---|---|---|
| GF-split-Gal4 | R17A04-p65ADZp (attP40); R68A06-ZpGdbd (attP2) | [38] |
| SS1540-split-Gal4 | VT023490-p65ADZp (attP40); R38F04-ZpGAL4DBD (attP2) | [47] |
| UAS-CsChrimson | 20XUAS-CsChrimson-mVenus (attP18) | [46] |
| CSMH | Canton S wild type | Martin Heisenberg, University of Wurzburg |

eclosion. Flies were reared at 25˚C. All experiments were performed on 2 to 3-day old male and female flies between the times of 12:00 and 20:00. Genotypes are as listed in Table 1.

*Drosophila* **tethering.** *Drosophila* were anesthetized in a conical tube on ice for approximately 10 minutes. Single anesthetized flies were then tethered to one end of a 0.1 mm tungsten wire with UV glue under a stereoscope, using a small paintbrush to position and restrain the fly. The other end of the tungsten wire was inserted into clay for stability and future positioning. Flies were then placed in covered boxes to recover in the darkness for more than five minutes prior to the start of experiments. Flies were kept in covered boxes between experiments to avoid unintended light stimulation.

**Experimental protocol.** For each experiment, a single tethered, dark recovered fly was placed in the center of the stereoscope objective, 2 mm in front of the LED. The students next attached their smartphone to the camera mount on the stereoscope using double sided tape. The stereoscope was then used to focus the smartphone camera on the fly via a slow-motion video app (SloPro, https://apps.apple.com/us/app/slopro/id507232505). We chose the SloPro app because it uses optic flow to simulate 1000 frames per second (fps) high speed video capture. This permits students to recognize the initiation or cessation of evoked behaviors that are quite rapid and difficult to observe by the human eye and not be limited by the highest frame rate of their smartphone camera. We also identified a slow-motion app for android phones (Slow Motion Video FX, https://m.apkpure.com/slow-motion-video-fx/com.mobile.bizo.slowmotion) that is sufficient for behavior annotations. We anticipate, with the continuous development in smartphone video technology (the newest iPhones and Android phones are capable of 240 fps at 1080p and 960 fps at 720p, respectively) a high speed "simulation" app will not be required in the near future.

Tethered flies produced "grooming" and "struggling" leg movements which were ideal to precede activation of freezing behavior but were not ideal for annotating takeoff behaviors. For giant fiber (GF) activation experiments, experimental and control flies were provided a small piece of a Kimwipe (Kimberly-Clark) to "kick" away during a takeoff. For freezing experiments, Kimwipes were provided to and then removed from flies to induce an active state if the fly appeared to be immobile. Once the fly was positioned, the smartphone camera was adjusted, and the fly was in the ideal behavioral state, a video recording and a 1.5 second delayed 300 ms red light stimulation were initiated by touching the record button on the smartphone. After the light stimulation, the recording was ended by again clicking the record button. Each fly was exposed to only one light stimulation.

**Data analysis.** To validate the device and ensure evoked behaviors were both obvious and quantifiable before introducing the device to high school students, we annotated fly escapes across experimental and control videos as the first frame of the middle leg extension (takeoff) that occurs before flight initiation. We annotated freezing behavior across experimental and control videos as every frame that consisted of a consecutive 120 ms absence of leg motion after the cessation of the light stimulation. The annotations were performed by an experimenter that was not blinded to the genotypes.

**Statistics.**   Statistical analyses were performed in MATLAB and SPSS. A Jarque-Bera test was first used to determine if data followed a normal distribution. If the data were not normally distributed, the appropriate non-parametric test was selected based on the number of sample groups. All statistical tests are stated in the figure captions.

## Optogenetics workshop for high school students

**Ethics.**   After utilizing the Drexel IRB's decision tool for human research determination and in consultation with IRB staff, it was determined that this project did not require IRB submission. Written participation consent was obtained and consent from parents or guardians was required for all minors.

**Workshop.**   After validating our device, we incorporated it into a neuroengineering workshop for high school students that was part of the immersive, week-long Drexel BIOMED Summer Academy (https://drexel.edu/biomed/resources/prospective-students/summer-academy/). The purpose of this workshop was to teach high school students neuroscience concepts and introduce them to emerging neurotechnologies through active learning. The 1.5-hour workshop was held 3 times, with 14–17 high school students attending each workshop.

Each workshop began with an initial assessment in the form of a short quiz. The quiz asked each student to answer the following four questions with short answers and sketches: (1) "What is a neuron?" (2) "What is a neural circuit?" (3) "What is a sensorimotor transformation?" (4) "What is optogenetics?". Following the quiz, the instructor introduced the fruit fly *Drosophila melanogaster* and the rationale for why the fly has become such a valuable model organism for investigating questions in neuroscience and developing neurotechnologies. The instructor utilized *Drosophila* takeoff escape and freezing behaviors to explain the concept of sensorimotor transformations and provided an overall reference for *Drosophila* central nervous system anatomy. Next, the instructor introduced optogenetics and led a discussion with the students on how optogenetics can be applied to discover neural circuits that underlie sensorimotor transformations. The instructor then segued into the experimental design for the active learning activity incorporating the optogenetics device.

The high school students were then divided into six groups of 2–3 students and provided a stereoscope, an optogenetics device, and a covered box of tethered flies. Before the workshop, students were asked to download the free SloPro App on their iPhone. They next positioned a tethered fly in the center of their stereoscope, and placed the optogenetic stimulation LED, mounted to a folded piece of cardboard, 2 mm away from the fly. They finally positioned the camera mount and adjusted the focus on the stereoscope to generate an in-focus image of the fly that enabled all appendages and wings to be clearly identified.

To run a single stimulation experiment, the students first turned on the board, as indicated by the green Board LED. They next pushed the camera shutter on their smartphone. This movement was captured by the motion sensor, which then initialized the stimulation, turning on the red optogenetics LED for 300 ms and indicating that a stimulation had occurred by switching the Board LED to blue. The students next ended the video capture by again clicking on the camera shutter. The students then waited until the Board LED returned to green before performing their next stimulation experiment.

For this workshop, we expressed CsChrimson within sensorimotor neurons in *Drosophila*. In *Drosophila*, descending neurons (DN) represent a bottleneck for the fly brain to communicate with motor centers in the ventral nerve cord (the fly version of the spinal cord). Here, we selectively drove expression in the GF (also known in the literature as DNp01), a pair of DN that trigger a stereotyped escape behavior through the tergotrochanteral muscle (middle leg

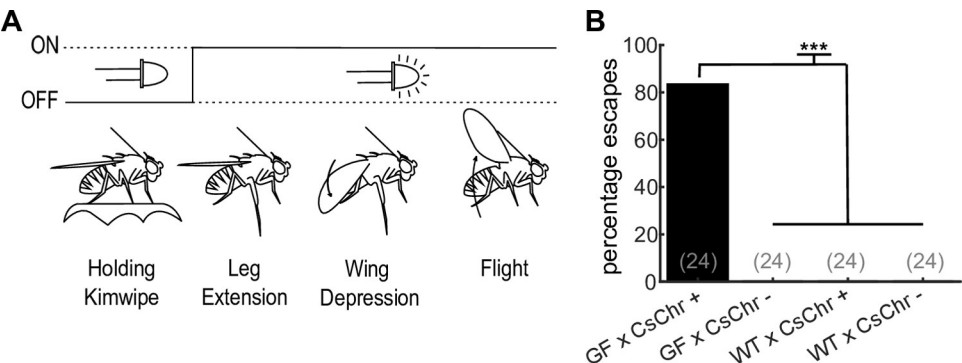

**Fig 2. The optogenetics device replicates prior GF activation studies.** (**A**) Annotated response of optogenetic GF activation shows the fly holding a small piece of a Kimwipe before light stimulation, followed by subsequent leg extension, wing depression, and flight initiation after light stimulus presentation. (**B**) Percentage of escapes across genotypes (n = flies as stated in figure, χ2 test, P << .001, Bonferroni correction post hoc, *** = p < .001). Abbreviations: retinal food (+), standard food (-), *GF-split-GAL4* (GF), *Canton S wild type* (WT), *UAS-CsChrimson* (CsChr).

extension for jumping) and dorsal lateral muscle (wing depression) (Fig 2A) [38, 48]. We also drove CsChrimson expression in a population of neurons that trigger freezing behavior (Fig 3A) [41, 44]. We found both escape and freezing behaviors were easily detected by our novice high school students.

Each student group was provided six flies expressing CsChrimson in the GF and six flies expressing CsChrimson in freezing neurons. All students were blinded to the genotypes and were only informed that the genotypes contained CsChrimson expressing neurons that may play a role in generating avoidance/escape behaviors. The student's goal, across these experiments, was to determine whether each genotype contained neurons that drove an observable behavior and, if so, to describe the particular behavior generated by the neurons. Flies were kept within the covered box between experiments. Students were also provided small pieces of Kimwipe to give to their flies.

After performing an experiment, the students then annotated the fly's behavior by reviewing the video on their phone and noting the time of escape or freezing events. Middle leg extension and flight initiation were classified as escape behaviors, and a cessation of fly motion

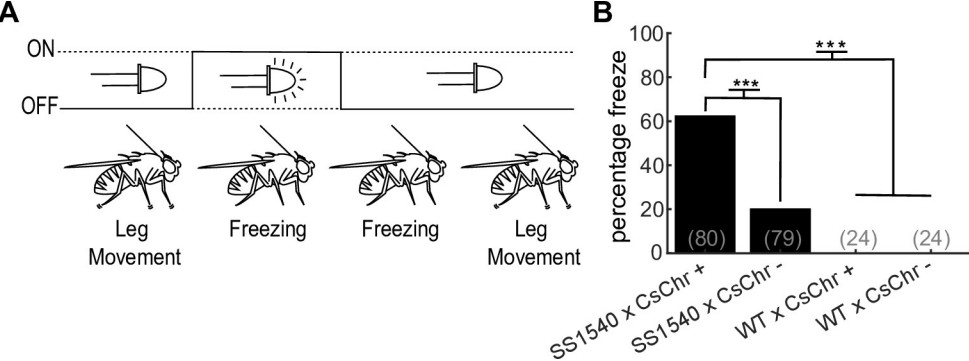

**Fig 3. The optogenetics device replicates prior freezing neuron activation studies.** (**A**) Annotated response of optogenetic activation of the freezing line shows the fly moving its legs before light stimulation, freezing during and after light stimulation, and then returning to leg motion (**B**) The percentage of flies that freeze with light stimulation (n = flies as stated in figure, χ2 test, P << .001, Bonferroni correction post hoc, *** = p < .001). Abbreviations: retinal food (+), standard food (-), *SS1540-split-Gal4* (SS1540) *CSMH wild type* (WT), *UAS-CsChrimson* (CsChr).

for more than 120 ms was classified as a freezing behavior. After all data were analyzed, the students regrouped with the instructor to discuss their results and made postulations on how activated cell types may contribute to *Drosophila* collision avoidance behaviors.

At the end of the workshop, the students were provided the same quiz they received at the beginning of the workshop. All quiz results were graded using a 4-point scale, with 0 representing no understanding of the concept, 1 representing novice understanding of the concept, 2 representing moderate understanding of the concept, and 3 representing full understanding of the concept.

## Results

### Device validation–activating escape neurons

Prior research has demonstrated that the GF, a bilateral pair of descending neurons, drive takeoff escape behaviors with optogenetic activation [38]. Light evoked takeoff escapes are performed with a high probability and are easy to classify with minimal training, as they contain a rapid, synchronized middle leg extension, wing depression, and initiation of flight (Fig 2A). To determine whether our optogenetics device could reproduce these results, we activated tethered flies and imaged their behavior. We expressed a channelrhodopsin CsChrimson [46] that is activated by red light in the GF using a highly specific genetic driver line *GF-split-Gal4* [38]. Experimental flies were fed retinal, and controls consisted of retinal and non-retinal fed flies of the appropriate genetic backgrounds. We found that only flies that expressed CsChrimson in the GF and had been fed retinal displayed takeoff behaviors (Fig 2B and S1 Movie). In all, we demonstrated that our inexpensive device is able to replicate prior GF activation data.

### Device validation–activating freezing neurons

Prior research has determined that another driver line (*SS1540-split-Gal4*) contains neurons that generate freezing behaviors similar to those observed in natural fly behavior as an object approaches on a direct collision course [41, 43]. Freezing behaviors are performed with high probability and are easy to classify with minimal training, as they consist of a complete interruption of appendage movement (Fig 3A). The exact neurons that generate freezing behaviors, however, remain unknown. Freezing was initially attributed to a bilateral pair of descending neurons, called DNp09 [41], but more recently it has been suggested DNp09 instead drives forward locomotion, and that other labeled neurons in *SS1540-split-Gal4* are responsible for the freezing phenotype [44]. We therefore next investigated whether our low-cost device could replicate prior results by activating the neurons labeled by this line through CsChrimson expression. Experimental flies were fed retinal, and controls consisted of retinal and non-retinal fed flies of the appropriate genetic backgrounds. We found that flies expressing CsChrimson in neurons labeled by this line displayed freezing behavior at a high probability (Fig 3A, 3B and S2 Movie). Interestingly, we also witnessed a small (but significant) proportion of freezing behaviors in non-retinal fed flies (Fig 3B). As standard cornmeal/molasses medium may contain a trace amount of β-carotene that could enable a small amount of functional CsChrimson if converted to retinal, our results suggest the neurons that initiate freezing behavior may have a very low activation threshold.

### Incorporating our device in an optogenetics workshop for high school students

After validating our optogenetics device, we next incorporated it into neuroengineering workshops for high school students. To evaluate whether our workshops had achieved our

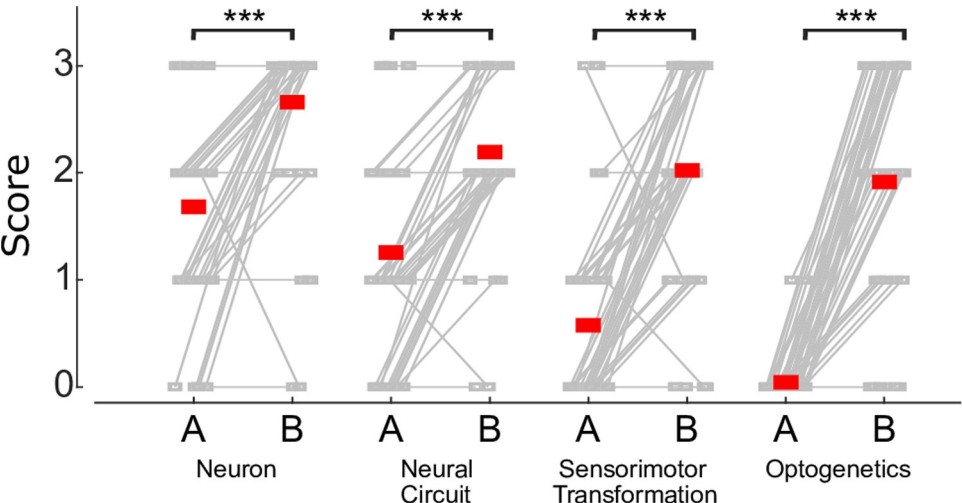

**Fig 4. Workshop assessment outcomes.** A total of 47 students distributed over three workshops were evaluated on four concepts, both before and after the workshop. Gray connected boxes indicate individual student scores and red squares indicate averages (Wilcoxon signed rank test, *** = p<<0.001). Abbreviations: quiz administered before the workshop (B); quiz administered after the workshop (A).

established learning outcomes, we assessed each student's ability to explain, with text and sketches, four neuroscience concepts by providing them with a short quiz before and after the workshop. We scored their ability to explain a concept using a four-point scale, with 0 representing no understanding of the concept, 1 representing novice understanding of the concept, 2 representing moderate understanding of the concept, and 3 representing full understanding of the concept. We compared the before and after workshop scores for each question and found that the students' ability to explain the four concepts (neuron, neural circuit, sensorimotor circuit, and optogenetics) significantly increased after participating in the workshop (Fig 4).

## Discussion

We have developed a novel, low-cost optogenetics device that enables one-button triggered red-light neuronal stimulation and synchronized video capture. The device can be modified to accommodate a variety of stereoscopes and can be used with a variety of smartphone models, greatly reducing the cost of high-speed cameras commonly used in optogenetics behavioral assays. We demonstrate this device is capable of reproducing optogenetics experiments performed with expensive equipment. We also demonstrate that incorporating this device into a high school neuroengineering workshop allows students to engage in active learning that greatly improves their understanding of neuroscience and neurotechnology concepts.

Our device is able to reproduce prior results indicating that optogenetic activation of the GF evokes takeoff escapes. Optogenetically induced GF mediated escapes consist of a characteristic short duration leg extension and wing depression (Fig 2A). Observed naturalistic *Drosophila* escapes can also include a long duration wing raise prior to leg extension that are driven by other descending neurons [38, 45]. Although escape preparation latency was not measured in this study (as we did not have the temporal resolution when using an app that simulated high speed video capture), we believe the continual increase in high frame rate smartphone video-capture technology (currently reaching 960 fps at 720p) should enable future experimenters to accurately and precisely annotate individual motor components that comprise these and other *Drosophila* behaviors.

Our device is also able to reproduce prior results indicating that optogenetic activation of *SS1540-split-GAL4* evokes freezing behavior [41, 44]. Interestingly, we did not observe the other reported behavioral output of this line, an increase in locomotion driven by DNp09 [44]. We postulate this is because our device is not designed to directly measure locomotion and a lack of tarsal contact for our tethered flies biases the behavior towards the freezing phenotype. We also find that a small but significant subset of non-retinal fed flies freeze in response to the light stimulus, a response that has not been previously reported. We hypothesized this may be due to low threshold neuron activation due to a trace amount of β-carotene in standard fly food. We leave investigating which neurons are responsible for freezing and the possibility for low threshold activation in these neurons to future researchers.

We demonstrate that our well characterized device can be successfully incorporated into a hands-on neuroengineering workshop for high school students. We find the workshop greatly improves the knowledge of participants across four key neuroscience/neurotechnology concepts. The demonstrated device utility and discoveries made when validating the device highlight the device's adoptability in high school labs: our device is low cost, made partially of recycled cardboard, and can be assembled by a tech savvy high school student with guidance from an instructor. Our device also overcomes a limitation of prior high school focused devices in that it enables the acquisition and quantification of behavioral data [27, 37].

We also highlight challenges associated with our workshop/device that may be improved upon with future renditions. At present, the workshop requires a partnership with a *Drosophila* research laboratory in order to obtain all-*trans*-retinal fly food, as there is no current commercial source. β-carotene has been used as an alternative to all-*trans*-retinal in *Drosophila* optogenetics experiments [49]. Adding carrot juice to fly food could therefore be a possible low-cost, easily accessible alternative but this has yet to be tested. Next, while we have found tethering under a stereoscope to be sufficient for a trained student, a newly designed, inexpensive fly tethering apparatus could be incorporated into the workshop to enable a novice to consistently tether flies [21]. Third, our device currently runs in open loop. Although this is appropriate for the behaviors we are characterizing, incorporating closed loop strategies implemented in other lower-cost devices may make our device useful in investigating a broad range of fly behaviors [20, 21]. Our device also requires a functioning stereoscope which may not be present in all high school labs. Future iterations of the device could incorporate a smartphone camera mount, 3-axis-manipulator, and a smartphone macro lens to eliminate the need for a stereoscope. Finally, our workshop assumes at least one high school student in every group of three will have a functioning smartphone. We incorporated smartphones into our device as they are currently the most ubiquitous technology for videography. To overcome financial barriers, however, smartphones could be replaced by high-speed cameras as the cost of the technology decreases. In all, we anticipate advancements in video technology will enable our device to be adopted in undergraduate and graduate research programs looking for low-cost alternatives.

## Supporting information

**S1 Fig. Stereoscope Camera Mount.**
(PDF)

**S2 Fig. Circuit diagram and connectivity diagram.**
(PDF)

**S1 Table. Materials with purchasing information.**
(PDF)

**S1 Movie. Escape example.**
(MP4)

**S2 Movie. Freeze example.**
(MP4)

**S1 File. Arduino code for optogenetics device.**
(ZIP)

## Acknowledgments

We would like to acknowledge HyoJong Jang, David Goodman, Kaylynn Curfman, and Dolores Conover who helped facilitate the workshop.

## Author Contributions

**Conceptualization:** Catherine R. von Reyn.

**Data curation:** Liudi Luo.

**Formal analysis:** Liudi Luo.

**Funding acquisition:** Catherine R. von Reyn.

**Investigation:** Liudi Luo, Bryce W. Hina, Brennan W. McFarland, Jillian C. Saunders, Natalie Smolin.

**Methodology:** Liudi Luo, Bryce W. Hina, Brennan W. McFarland, Jillian C. Saunders, Natalie Smolin, Catherine R. von Reyn.

**Project administration:** Catherine R. von Reyn.

**Software:** Liudi Luo, Bryce W. Hina.

**Supervision:** Catherine R. von Reyn.

**Validation:** Liudi Luo, Bryce W. Hina, Brennan W. McFarland, Jillian C. Saunders, Natalie Smolin.

**Visualization:** Liudi Luo, Bryce W. Hina, Catherine R. von Reyn.

**Writing – original draft:** Liudi Luo, Bryce W. Hina, Brennan W. McFarland, Jillian C. Saunders, Natalie Smolin, Catherine R. von Reyn.

**Writing – review & editing:** Liudi Luo, Bryce W. Hina, Brennan W. McFarland, Jillian C. Saunders, Natalie Smolin, Catherine R. von Reyn.

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
