## [Decision Letter · Decision Letter 0]

14 Dec 2021

PONE-D-21-29142Introducing neurotechnology with an optogenetics device and workshop for high school studentsPLOS ONE

Dear Dr. von Reyn,

Thank you for submitting your manuscript to PLOS ONE. After careful consideration, we feel that it has merit but does not fully meet PLOS ONE’s publication criteria as it currently stands. Therefore, we invite you to submit a revised version of the manuscript that addresses the points raised during the review process.

 Specifically, we ask you to integrate the recommendations made the three reviewers: Provide a better description of the existing literature and acknowledge previous work done to bring Drosophila neuro-engineering to high schools . Add missing references. Soften unnecessary or unjustified claims about novelty (requested).Complete missing information related to the methodology (requested). Add a bill of material (table) with the reagents and providers (requested).Discuss alternatives to the use of a smartphone and stereoscope (recommended).Ensure that the cell-phone application works for both iPhone and Android phones (strongly recommended). In addition, please consider the following requests and suggestions made by specific reviewers: 

 Reviewer 1:Improvement of layout (arrow-based representations) and information represented in the figures (strong recommendation). Improvement of the statistics and potential increase of sample sizes (Figure 4) or a better justification of the appropriateness of the statical tests used (requested). Discuss how high school students can obtain trans-retinal. Discuss ways to mitigate potential barrier in the use of the tool in an educational context (strong recommendation).Reviewer 2:Provide details about the automatic triggering of the Arduino and its accuracy (strong recommendation).Add presentation of raw data in Figure 4 (strong recommendation). Reviewer 3: Alternative to UV glue in high schools where UV light sources might not be available (strong recommendation). 

We look forward to receiving your revised manuscript.

Kind regards,

Matthieu Louis

Academic Editor

PLOS ONE

Journal Requirements:

Reviewers' comments:

Reviewer's Responses to Questions

**Comments to the Author**

1. Does the manuscript report a protocol which is of utility to the research community and adds value to the published literature?

Reviewer #1: Yes

Reviewer #2: Yes

Reviewer #3: Yes

2. Has the protocol been described in sufficient detail?

Descriptions of methods and reagents contained in the step-by-step protocol should be reported in sufficient detail for another researcher to reproduce all experiments and analyses. The protocol should describe the appropriate controls, sample sizes and replication needed to ensure that the data are robust and reproducible.

Reviewer #1: Partly

Reviewer #2: Partly

Reviewer #3: Yes

3. Does the protocol describe a validated method?

Reviewer #1: Yes

Reviewer #2: Yes

Reviewer #3: Yes

4. If the manuscript contains new data, have the authors made this data fully available?

Reviewer #1: Yes

Reviewer #2: Yes

Reviewer #3: Yes

**5. Is the article presented in an intelligible fashion and written in standard English?**

Reviewer #1: Yes

Reviewer #2: Yes

Reviewer #3: Yes

6. Review Comments to the Author

Reviewer #1: Introducing neurotechnology with an optogenetics device and workshop for high school students

This work describes the development of a low-cost optogenetic equipment used to educate high school students and the curriculum around it. The simple low-cost set up and the workshop materials are highly attractive.

Major comments:

Fig. 1E: The arrows are confusing if not misleading. For example, Ultrasound module has an arrow to the status indication. But the Ultrasound module does not control it. Another example is the Board LED module. This has an arrow to the Camera mount. But the board LED does not control or manipulate the camera mount. Also, what are the switch module and the stereoscope LED module (although I can guess)? They are not explained in anywhere in the text. I strongly suggest revising this figure and clearly explaining every component and arrows.

In Figure 4, performing statistical test on sample size 3 is not well justified. I suggest collecting at least 5 samples (desirably 10 samples or more) for any statistical tests. If this is not feasible, consider other ways of testing the effectiveness. (E.g., instead of using the average point of each session, individual students’ points can be tested using 2-way Anova for each question, which should provide more believable statistics. Or even three-way Anova may be considered for the entire dataset.)

I suggest including the actual light power of the 627 nm LEDs measured at 2 mm distance. Even though high school students may not need to know the details, the reader of the paper may want to know the technical specs of the system.

Related, wouldn’t a regular red LED (not a high power LED) activate GF?

In the Introduction & Discussion: Low-cost devices to perform optogenetics or behavioral experiments with Drosophila have been developed in multiple labs. (Below are such examples.) These different approaches need to be compared and discussed. For example, the setup of the current manuscript is even more cost-effective and easier to implement, which can be emphasized. Or, the current setup can be implemented by tech-savvy high school students without the help of instructors.

- The 100 Euro Lab: A 3D-printable open-source platform… PLOS Biology, Chagas et al., 2017

- PiVR: An affordable and versatile closed-loop platform… PLOS Biology, Tadres and Louis, 2020

- An inexpensive, High-Precision, Modular Spherical Treadmill Setup… Frontiers in Behavioral Neuroscience, Loesche and Reiser, 2021

In Discussion: On a side note, I believe that fly genetics and optogenetics can bring a huge impact on K-12 education. There is a potential to renovate the biology curriculum in K-12 schools. However, the biggest barrier to fly optogenetic experiments in general public is to find the all-trans-retinal (ATR) food, which requires purchasing ATR powder, dissolving it in ethanol, and keeping it in a low temperature until mixed in the regular food. All of these steps are close to impossible for high school students to do by themselves. In most cases, such food can only be obtained via fly labs in a nearby university. As a result, most students cannot perform exciting optogenetic experiments at home for an extended period time, which is essential to develop scientific insights. Although this may be well beyond the scope of this study and not necessary for this study, it would be appreciated if authors share any idea about, if any, or discuss how to mitigate this major barrier.

Minor comments:

The fly and the red LED are a bit small Fig. 1C compared to other figures in Fig. 1. You may want to enlarge it.

Fig. 1C: The fly is tethered at the tip of a wire. I suspect that the wire might be very wobbly. How do you stabilize it?

Line 119: typo?: It is consisted of

Using a cardboard to mount a phone is a great idea to reduce the cost because most phone mounts are at least $15. But the proposed design seems to work only with the Leica EZ scope. The phone mounting part and the part covering the eyepiece could be separated in the supplementary figure and you can offer a multiple designs or instructions to build for other brands of stereoscopes.

Line 134: Wiring diagram uses a prototyping breadboard, but in Fig. 1, a shield is used. Although this is a very minor issue, it may be a good idea to clarify this for those who are not familiar with the Arduino ecosystem.

Line 172-173: I assume that the turning on the 627nm LED is delayed for some amount of time (e.g., 3 seconds) to wait for the phone camera to start recording. Please clarify.

Line 194: Anesthetization of flies on ice for 10 minutes may easily kill the fly. Authors may have used a method to prevent hemolymph from freezing. That is, the temperature may have been higher than 0 C. Please clarify. (E.g., A vial with fly may be place in an ice bucket for a couple of minutes, then can be transferred to a plate with a temperature at around 2 C.)

Line 195: Was the gluing done under a microscope? Was the fly constrained in a sarcophagus?

Line 197: Why do you place the fly in a freezer box (below 0 C)? Or is it just a Styrofoam box without ice?

Line 202, 264: Does not have to be an iPhone. Or does it?

Line 217: Placing the Kimwipe for the fly to hold (and its purpose) should be mentioned before this line.

Line 224: Please describe how to quantify the freezing if the fly was not moving before the light ON.

Overall, the Methods section requires more attention to details.

Reviewer #2: I would like to thank the authors for putting this manuscript together. In the interest of keeping my comments easy to access, I uploaded them as a separate file.

Reviewer #3: Review of Luo et al.

Overview

This paper describes a new method for deploying Drosophila optogenetics in high school classrooms. Novel features include the use of cell phones to trigger Arduino controlled light sources and high-speed videography of behaving flies. The authors show the feasibility of their system and test drive it in a classroom setting. This work could provide an important new resource for educators but some revisions are necessary.

Major comments

The authors make strong claims about the novelty of bringing inexpensive tools for Drosophila optogenetics and neurotechnology more generally into classrooms. There have been quite a few efforts to do this and the authors could do a better job of acknowledging this body of work and then soften some of their claims accordingly.

In particular, there have been quite a few publications led by academics that outline the use of Drosophila neurogenetics and inexpensive tools for optogenetics in classrooms. There have also been strong efforts to bring neurotechnology in the form of neurophysiology into classrooms led by Backyard Brains. It would be nice to see sections added into the introduction and discussion that comment on how the work presented here extends beyond these efforts into new domains. The use of motion sensors to trigger LEDs and the use of a ‘Slomo’ app for high-speed videography for example, opens up fantastic opportunities for studying fast behaviours. Below is a list of articles that I would suggest reviewing and considering for citation at various points in introduction and/or discussion.

• Pulver et al., 2011 (first report of inexpensive Drosophila optogenetics methods in classrooms) https://journals.physiology.org/doi/full/10.1152/advan.00125.2010

• Marzullo et al., 2012 (presentation of inexpensive amplifier now widely used in education https://www.ncbi.nlm.nih.gov/pmc/articles/PMC3310049/

• Pulver and Berni, 2012 (reviews of use of Drosophila neurogenetics in teaching labs)

https://www.ncbi.nlm.nih.gov/pmc/articles/PMC3592735/

• Titlow et al., 2014a,b (reports of deployment of Drosophila optogenetics in high schools) https://uknowledge.uky.edu/cgi/viewcontent.cgi?article=1081&context=biology_facpub and https://f1000research.com/articles/6-117/v1

• Michels et al., 2017 (multi-lingual resources for deploying Drosophila neurogenetics in secondary school classrooms) https://www.ncbi.nlm.nih.gov/pmc/articles/PMC5395560/

• Chagas et al., 2017 (inexpensive lab for optogenetics) https://www.ncbi.nlm.nih.gov/pmc/articles/PMC5515398/

• Rhodes, unpublished (online SFN training resources for educators) https://neuronline.sfn.org/training/module-5-implementing-optogenetics-in-the-classroom

• Villinsky et al., 2018 (refinement of optogenetics exercises, including inexpensive methods for activation of giant fiber system in adult flies) https://pubmed.ncbi.nlm.nih.gov/30254546/

• Tadres and Louis, 2020 (inexpensive system for closed loop optogenetics experiments)

https://journals.plos.org/plosbiology/article?id=10.1371/journal.pbio.3000712

This is not meant to be an complete list of all relevant work; there are certainly other sources that could be cited. Just to be clear, in this type of report, I would of course, not expect an exhaustive review of the entire field, but think it is appropriate to include a few lines acknowledging efforts to make specific relevant types of neurotechnologies more accessible to educators, as this is a core theme of the paper.

Minor comments

Title

Suggest modulating to emphasize use of high speed videography to distinguish this work from other attempts to bring optogenetics and neurotechnology generally into classrooms.

Abstract

The abstract makes a very strong initial claim to the effect that neurotechnology is not being brought into high school classrooms, when in fact there are quite a few ongoing efforts to do this (see comments on intro and discussion). Suggest softening this initial claim to be inclusive of ongoing efforts to bring neuroscience technologies into classrooms. Bringing cutting edge neuroscience technologies into high school classrooms can be perhaps be presented as an ongoing challenge rather than as a current ‘failure to follow suit’.

Suggest acknowledging that Drosophila optogenetics is now beginning to be deployed in both undergraduate and high school classrooms, but then pointing out the lack of tools for quantitatively studying high speed behaviours. This would then provide a nice rationale and underscore the specific value of this study.

The type of optogenetics experiments performed are not well defined in the abstract. Suggest modulating abstract to report on the actual experiments done (i.e. fast escape behaviours in adult flies). This will help readers understand the scope of the experiments presented.

Introduction

Line 48: The authors state:

“In recognition of this trend, universities have significantly increased their offerings of undergraduate courses in neuroscience and neuroengineering (6). High school curricula, however, have not followed suit (7, 8).”

Suggest softening this statement to acknowledge previous attempts to bring neurotechnology into high schools. I would suggest citing the work of Backyard Brains, Cooper lab at U. of Kentucky, and Gerber lab at the Leibniz Institute for Neurobiology, for specific examples of efforts to include neurotechnology and neurogenetics into high school curricula.

Line 62: Suggest softening this claim to acknowledge previous work.

Line 68: The statement ‘However, most cutting-edge neuroscience/neurotechnology experiments are costly and unrealistic to replicate in a high school setting (18).’ is not accurate. Again, suggest narrowing this claim to acknowledge previous work. Authors could make the point that although progress has been made in this domain, there is much work still to be done!

Line 73-86: This paragraph should be rewritten to acknowledge previous efforts to build and test inexpensive systems for Drosophila optogenetics. In particular, it is not accurate to claim that there are no inexpensive systems for performing optogenetics experiments in Drosophila. There are actually quite a few available at this point (see ref list above). It is true though that there is a need for new systems, especially those that enable quantitative analysis of high speed behaviours in classrooms.

Methods

General: The supplemental materials are solid, but suggest including a simple table that has sources for all components and materials needed. Suggest also reviewing methods to ensure sources of materials and reagents are documented appropriately in text.

Line 186: 'Retinal’ should be changed to ‘all-trans-retinal’ to reduce confusion for readers.

Line 195: Use of UV glue necessitates access to a directable UV light source, which could be problematic for high school teachers. Suggest commenting on possible alternatives methods.

Line 296: Suggest delineating how exactly students annotated behaviours – did they simply review on their phones and note times of events manually?

Results

Line 317: suggest checking to see if GF driver line used was same as in Villinsky et al., 2018. If so suggest citing that paper.

Discussion:

General: The discussion is well written, but it could be improved by addition of a section that compares and contrasts this work with other efforts to perform optogenetics experiments inexpensively, especially in adult flies. An honest presentation of the pros and cons of this approach compared to other systems would really enrich the discussion and help guide educators towards the right solution for their own particular classes.

Line 393: iPhones are quite expensive and beyond the budget of many students and teachers. Suggest commenting here explicitly on performance and suitability of less expensive phones and imaging devices.

Line 401: This may be a good place to comment on the challenges associated with working with tethered adult flies. Are there for example variants of these experiments that do not require UV glue and tethering?

Line 404: check for consistency of spelling of ‘retinal’ throughout manuscript.

7. PLOS authors have the option to publish the peer review history of their article (what does this mean?). If published, this will include your full peer review and any attached files.

Reviewer #1: No

Reviewer #2: **Yes: **Andre Maia Chagas

Reviewer #3: No

---

## [Author Response · Author response to Decision Letter 0]

15 Feb 2022

We have attached our responses as a .pdf which should be easier to read/follow.

---

## [Decision Letter · Decision Letter 1]

18 Apr 2022

An optogenetics device with smartphone video capture to introduce neurotechnology and systems neuroscience to high school students

PONE-D-21-29142R1

Dear Dr. von Reyn,

We’re pleased to inform you that your manuscript has been judged scientifically suitable for publication and will be formally accepted for publication once it meets all outstanding technical requirements.

Kind regards,

Matthieu Louis

Academic Editor

PLOS ONE

Additional Editor Comments (optional):

Reviewers' comments:

Reviewer's Responses to Questions

**Comments to the Author**

1. Does the manuscript report a protocol which is of utility to the research community and adds value to the published literature?

Reviewer #1: Yes

Reviewer #2: Yes

Reviewer #3: Yes

2. Has the protocol been described in sufficient detail?

Descriptions of methods and reagents contained in the step-by-step protocol should be reported in sufficient detail for another researcher to reproduce all experiments and analyses. The protocol should describe the appropriate controls, sample sizes and replication needed to ensure that the data are robust and reproducible.

Reviewer #1: Yes

Reviewer #2: Yes

Reviewer #3: Yes

3. Does the protocol describe a validated method?

Reviewer #1: Yes

Reviewer #2: Yes

Reviewer #3: Yes

4. If the manuscript contains new data, have the authors made this data fully available?

Reviewer #1: Yes

Reviewer #2: Yes

Reviewer #3: Yes

**5. Is the article presented in an intelligible fashion and written in standard English?**

Reviewer #1: Yes

Reviewer #2: Yes

Reviewer #3: Yes

6. Review Comments to the Author

Reviewer #1: All concerns have been addressed in this revision.

Just for information, UV torches are cheap at Amazon and work well with most UV glue.

Reviewer #2: I would like to thank the authors for updating their manuscript and addressing my previous questions and suggestions. The current version of the manuscript is good for publication.

Reviewer #3: The authors have made very thoughtful and thorough revisions in response to reviewer comments. All of my concerns have been addressed.

7. PLOS authors have the option to publish the peer review history of their article (what does this mean?). If published, this will include your full peer review and any attached files.

Reviewer #1: No

Reviewer #2: **Yes: **Andre Maia Chagas

Reviewer #3: No

---

## [Editor Report · Acceptance letter]

28 Apr 2022

PONE-D-21-29142R1 

An optogenetics device with smartphone video capture to introduce neurotechnology and systems neuroscience to high school students 

Dear Dr. von Reyn:

I'm pleased to inform you that your manuscript has been deemed suitable for publication in PLOS ONE. Congratulations! Your manuscript is now with our production department. 

Kind regards, 

on behalf of

Dr Matthieu Louis 

Academic Editor

PLOS ONE